# Canaloplasty in Pseudoexfoliation Glaucoma. Can It Still Be Considered a Good Choice?

**DOI:** 10.3390/jcm11092532

**Published:** 2022-04-30

**Authors:** Paolo Brusini, Veronica Papa, Marco Zeppieri

**Affiliations:** 1Department of Ophthalmology, Policlinico Città di Udine, Viale Venezia 410, 33100 Udine, Italy; papa.veronica87@gmail.com; 2Department of Ophthalmology, University Hospital of Udine, 33100 Udine, Italy; markzeppieri@hotmail.com

**Keywords:** canaloplasty, non-perforating surgical procedures, pseudoexfoliation glaucoma (PEXG), Schlemm’s canal, intraocular pressure (IOP)

## Abstract

Purpose: The aim of this study was to assess the long-term outcomes of canaloplasty surgery in pseudoexfoliation glaucoma (PEXG) patients. Material and Methods: A total of 116 PEXG patients with an intraocular pressure (IOP) > 21 mm/Hg and maximum tolerated local medical therapy who underwent canaloplasty from February 2008 to January 2022 were considered. Every six months, all subjects underwent a complete ophthalmic examination. The period of follow-up ranged from 2 to 167 months. Inclusion criteria included only patients for whom the entire procedure could be completed with a follow-up of at least 2 years. Results: Amongst the 116 PEXG patients, the entire procedure could not be performed in 10 eyes (8.6%), and thus they were not considered in the analysis. Twenty-three patients did not reach the two-year follow-up and another 16 patients during this time period were lost. A total of 67 patients with a mean follow-up of 49 ± 32.3 months were considered in the analysis. The pre-operative mean IOP was 31.2 ± 8.7 mm/Hg (range 20–60). The mean IOP at the two-year follow-up was 17.2 ± 6.7 mmHg, with a mean reduction from baseline of 44.9%. After two years, the qualified success rates according to three different criteria (IOP ≤ 21, ≤18 and ≤16 mmHg) were 80.6%, 73.1% and 61.0%, respectively. The total number of medications used pre- and at the follow-up at 2 years was 3.5 ± 0.8 and 1.2 ± 1.4, respectively. Early complications included: hyphema, in about 30% of cases; Descemet membrane detachment (4.9%); and IOP spikes > 10 mmHg (9.7%). A late failure with an acute IOP rise of up to 50 mmHg was observed in 41 cases (61.2%) after 3 to 72 months. Conclusions: Long-term post-operative outcomes of canaloplasty in PEXG patients appear to be quite good on average; however, an acute rise in IOP can be observed in more than 60% of the cases after a long period of satisfactory IOP control. For this reason, canaloplasty may not be suitable in eyes with PEXG, especially in patients with severe functional damage.

## 1. Introduction

Pseudoexfoliation glaucoma (PEXG) is a frequent form of secondary glaucoma due to deposits of fibrillary material in the juxtacanalicular portion of the trabecular meshwork [1]. It is known that PEXG is more aggressive than primary open-angle glaucoma (POAG) and scarcely responsive to medical treatment. Intraocular pressure (IOP) is usually higher and can show elevated spikes in eyes with PEXG compared to POAG, which may lead to a quicker progression of glaucomatous damage. Trabeculectomy using intra-operative antimetabolites remains the gold standard procedure in PEXG [2,3], even if the success rate seems to be lower in comparison with POAG [4,5]. Trabeculectomy is quite easy to perform and effective in reducing IOP; however, several late and early potentially serious complications can arise, such as hypotonus, atalamia, bleb infection, choroidal detachment, etc. Moreover, the scarring of conjunctival tissues, despite the use of antimetabolite drugs, often leads to a complete failure of this filtering operation over time.

Canaloplasty is a blebless, non-perforating technique, which became popular several years ago and involves the positioning and tensioning of a 10-0 prolene suture within Schlemm’s canal, which is previously dilated using a viscoelastic agent. This surgical technique can facilitate aqueous outflow through the natural pathways [6,7,8,9,10,11,12,13]. The main indications for canaloplasty include POAG, juvenile glaucoma and pigmentary glaucoma. Even if PEXG is generally considered a good indication for canaloplasty, very few studies have specifically addressed this issue [14,15,16].

The aim of this paper is to evaluate the long-term outcomes and complications of canaloplasty in a group of PEXG patients.

## 2. Materials and Methods

The investigation was based on a retrospective, single-surgeon, observational, non-randomized study of patients with PEXG. One-hundred-and-sixteen eyes from 116 patients with uncontrolled pseudoexfoliation glaucoma under maximum tolerated medical therapy with significant visual field damage progression underwent canaloplasty under local anesthesia. Surgery was performed by the same surgeon (P.B.) in multi-subspecialty ophthalmic departments, either at the Department of Ophthalmology in the Azienda Ospedaliero-Universitaria “Santa Maria della Misericordia” Hospital or the Department of Ophthalmology in Policlinico “Città di Udine”, in Udine, Italy, from February 2008 to January 2022.

The investigation was performed in accordance with the tenets of the Declaration of Helsinki and informed consent was obtained from all participants before surgery. The study was in compliance with institutional review boards (IRBs) and the HIPAA requirements of both clinics.

### 2.1. Inclusion Criteria

Inclusion criteria for this cohort included: patients diagnosed with PEXG having an IOP ≥ 20 mmHg with maximum tolerated medical therapy, typical optic nerve alterations and functional loss (based on the Glaucoma Staging System 2 (GSS2), ranging from early to moderate GSS2 stages 1–3) [17]. Visual fields had to show significant progression of defects in 2 consecutive tests assessed with the Guided Progression Analysis 2 (Carl Zeiss Meditec Inc., Dublin, CA, USA) program. Patients who underwent previous ocular surgeries (with the exception of cataract surgery) were excluded. Patients with narrow-angled eyes, other serious eye diseases and unwillingness to undergo surgery were also excluded. All patients in the analysis were older than 18 years.

### 2.2. Surgical Technique

All surgeries were performed under local anesthesia. Canaloplasty is widely used and well-reported in the current literature [6,12]. Briefly, this surgery commences with a conjunctival fornix-based flap and a 3 × 4 mm superficial scleral flap that is dissected forward by 1.5 mm into the clear cornea. Surgery continues with the creation of a deep scleral flap used to open Schlemm’s canal. This flap is then removed. The exposed 2 ostia of the canal are dilated using hyaluronic acid of high molecular weight (Healon GV, Johnson & Johnson Surgical Vision, Inc., Santa Ana, CA, USA). A special 200-micron microcatheter is used which is connected to a flickering red light laser source, useful for easy identification through the sclera of the distal tip (Nova Eye Medical Limited, Fremont, CA, USA). The tip is inserted within Schlemm’s canal and pushed forward for the whole 360° until it comes out of the other end. A 10-0 double prolene suture is then tied to the distal tip and the microcatheter is pulled back and withdrawn in the opposite direction from the canal. A small amount of viscoelastic agent is delivered during this step in Schlemm’s canal every two hours of circumference using a special screw-driven syringe. Surgery then involves knotting the suture under tension to inwardly distend the trabecular meshwork. Using 5 to 7 10-0 vicryl sutures, the superficial scleral flap is then sutured to provide a closure that is watertight to avoid any bleb formation. Then, 8-0 vicryl sutures are used to close the conjunctival flap to complete the surgery.

### 2.3. Main Outcome Measures

Every 6 months, all patients underwent a complete ophthalmic examination that included slit lamp examination, Goldmann applanation tonometry IOP measurement, fundus examination using a 78 D Volk lens, best corrected visual acuity (BCVA) with visual field testing (Humphrey Field Analyzer 24-2 SITA standard test) and gonioscopy.

The definition of success was based on three different criteria: post-operative IOP ≤ 21 mmHg, ≤18 mmHg and ≤16 mmHg, without any medical treatment (“complete success”) or with or without medical treatment (“qualified success”). The number of local medications taken before and after canaloplasty and the early and late complications were also taken into consideration.

In order to assess the long-term outcomes of canaloplasty, only patients with a minimum follow-up of 2 years for whom the full technique was successfully completed were taken into consideration.

## 3. Results

The whole standard surgical technique of canaloplasty could not be performed in 10 eyes (8.6%) due to the impossibility of cannulating the entire 360° of Schlemm’s canal. In these cases, surgery was converted either in viscocanalostomy, which was carried out by injecting viscoelastic agent up to the intracanalicular obstacle, or in deep sclerectomy, whereby two nylon 10-0 stiches were used to suture the superficial scleral flap. These eyes were not included in the analysis. Twenty-three patients did not reach the two-year follow-up and another 16 patients were lost during follow-ups. A total of 67 patients (33 woman and 34 men) met the inclusion criteria and were considered in the analysis (mean age: 67.8 ± 12.5 years; range: 49 to 82 years). Six patients were treated with a combination of prostaglandin and timolol. Fifty-seven were using three to four topical medications (prostaglandin + timolol + dorzolamide + brimonidine) and four also used oral carbonic anhydrase inhibitors. Thirty-seven (55.2%) were pseudophakic. The best corrected visual acuity in decimal points ranged between 0.6 to 1.0 (mean 0.8). The optic disc showed glaucomatous cupping, ranging between 0.6 and 0.9 (mean cup/disc ratio 0.7). Visual field damage ranged between stage 1 and stage 3 of the Glaucoma Staging System, with a mean deviation ranging between −1.2 dB and −13.6 dB. The follow-up time ranged from 24 to 167 (mean 58.9 ± 28.8) months. The mean pre-operative IOP was 31.2 ± 8.7, ranging from 20 to 60 mmHg. After 24 months, the mean IOP was 17.2 ± 6.7 mmHg, with a reduction from baseline in mean IOP of 44.9%. The mean IOP values over a period of 7 years at various follow-up sessions are reported in the box plot diagram (Figure 1).

The scatter plot in Figure 2 shows the pre-operative IOP and post-operative IOP values after 2 years.

The qualified and complete success rates based on the three different IOP cut-offs after 2, 3 and 4 years are reported in Table 1 and Table 2.

The number of medications used pre- and at the 2, 3, and 4-year follow-ups were 3.5 ± 0.9, 1.2 ± 1.4, and 1.3 ± 1.3, and 1.9 ± 1.3 respectively. A Wilcoxon matched-pairs signed-rank test revealed statistically significant reductions at all time points (*p* < 0.001).

Gonioscopy at each follow-up confirmed that the prolene suture was still in the right position within Schlemm’s canal for the whole follow-up period, with the exception of one eye in which the tensioned prolene caused suture cheese-wiring through trabecular meshwork after surgery without any further complications.

Post-operative complications occurring early (within 4 weeks after surgery) included: hyphema in 14 eyes (34.1%), which completely reabsorbed within one week; hypotonus (IOP < 5 mm/Hg) in one eye (2.4%), in which the IOP returned to normal values (16 mmHg) in a couple of weeks; detachment of Descemet membrane in 2 eyes (4.9%), which spontaneously reattached without the need to be drained in one month; and IOP spikes > 10 mmHg in 4 eyes (9.7%). In the latter cases, no medical treatment was added in order to reduce IOP, a part one case, where acetazolamide tablets to be taken three times/day were prescribed for a few days. In all these four eyes, the IOP spontaneously dropped under 18 mmHg after about a month. A transient visual acuity decrease was reported in several patients within a few weeks after surgery which was brought on by induced according to-the-rule astigmatism that tended to disappear within one month. A late failure with an abrupt IOP rise, with values ranging between 26 and 50 mmHg, was observed after 3 to 72 months in 41 cases (61.2%). The number of these elevated IOP cases observed during the follow-up is reported in Figure 3. In 17 cases, it was possible to control IOP either with medical treatment or with selective laser trabeculoplasty, while in 22 eyes a trabeculectomy using the previous scleral flap was performed with good results in 18 cases (81.8%). In one case, we performed an ab interno trabeculotomy, stripping the prolene suture under gonioscopic control. In another case, a diode laser cyclophotocoagulation was performed. These last two patients were well controlled with medical therapy.

## 4. Discussion

Surgery is often required to reduce ocular hypertension and limit damage progression in PEXG patients, especially considering that functional damage can be rapid and severe in PEXG patients [18,19,20]. In eyes with advanced visual field loss, very low post-operative IOP values are needed to preserve the remnants of vision. Only filtering procedures, such as trabeculectomy or ExPress implant using antimetabolites, can offer these low IOP values and should be the preferred choice in these patients [21].

In selected patients who show mild to moderate functional damage, however, minimally invasive glaucoma surgery (MIGS) techniques, such as i-Stent implant [22], gonioscopy-assisted transluminal trabeculotomy [23,24] and XEN gel implant [25,26], can be taken into consideration. Non-perforating techniques, such as deep sclerectomy [27,28,29,30] or canaloplasty, may be an interesting alternative, considering the higher hypotensive efficacy when compared to MIGS and the lower rate of complications compared to trabeculectomy.

One of the main advantages of canaloplasty is that this type of surgery reduces IOP without requiring the formation of a filtering bleb [31]. For this reason, canaloplasty could be a viable option in selected patients having a high risk of conjunctival bleb failure with filtrating surgery, which is typically seen in eyes that have been treated with multiple local drop therapy for numerous years. This issue, however, needs further investigation considering the lack of studies in the current literature in this field. Another important advantage of this surgical option is the simplified follow-up and lower post-operative complication rates compared to the relatively high number of manipulations for blebs required after trabeculectomy (up to 78.2% of cases) [32].

The drawbacks of canaloplasty include the need for specific and expensive instrumentation and a steep learning curve. Another disadvantage, especially for beginners, is the proper cannulation of Schlemm’s canal, which can be difficult or not fully achieved in some cases. Canaloplasty, however, can be easily converted into a viscocanalostomy or a deep sclerectomy in these cases. In eyes that show mid-term failure after a successfully performed canaloplasty, a goniopuncture with YAG laser can be considered. In cases that do not show a sufficient reduction in IOP after canaloplasty and/or where medical therapy is not well tolerated or insufficient for lowering IOP, either a trabeculectomy or an implant of a drainage tube should be considered [33].

It is important to point out that in our cohort more than 60% of the PEXG eyes had an abrupt rise in IOP after years of satisfactory IOP stabilization. This long-term post-surgical complication, which is of utmost importance in these eyes at risk of functional progression, is poorly documented in the current literature [34]. According to our experience, this complication is more frequently observed two to four years after surgery.

The pathogenetic mechanisms behind such a late complication are not well known and need to be addressed in future studies. One possible reason could be related to the continuous production and accumulation of pseudoexfoliative material in the angle structures which can occlude the existing compromised aqueous humor outflow pathways after a short period of time, which may be due to the physiopathological mechanisms of the disease and by the effects of numerous years of medical drugs. This hypothesis is supported by the observation that a similar late IOP rise can be found only in 13.7% of POAG eyes (personal unpublished data obtained from a cohort of 117 POAG patients with a similar follow-up period who underwent canaloplasty performed by the same surgeon). The prolene suture inside Schlemm’s canal could also be involved in the scarring process leading to the increase in outflow resistance.

Future studies based on ultrasound biomicroscopy, preferably with 80 MHz transducers, or high-resolution anterior segment OCT [35,36], could help clarify, at least in part, the anatomical changes in Schlemm’s canal and in the trabecular meshwork in eyes showing long-term surgical failures. Histological studies conducted on human trabecular meshwork specimens could definitely provide a better comprehension behind the pathological post-operative induced structural changes in these eyes.

The onset of important IOP spikes can give rise to acute signs and symptoms in these patients, especially if IOP reaches high values. Urgent trabeculectomy can usually be effective in normalizing IOP in these situations, especially considering that the conjunctiva tends to in good condition after a long period without local medical treatment. Unfortunately, in some cases, this rise in IOP can be slower and less pronounced and can go totally unnoticed, leading to a worsening of the damage already present, which can cause a potentially devastating visual impairment. Based on these clinically important considerations and the post-surgical risks involved, all patients with PEXG who have undergone canaloplasty should be carefully managed and thoroughly monitored for life

Our study has several limitations, the most important being that it is based on retrospective results for a cohort of eyes and that a control group was not considered. The aim of our study, however, was not comparative in nature but to assess the long-term effectiveness of canaloplasty in pseudoexfoliation glaucoma, especially with regard to possible late complications of this procedure. The IOP cut-off values for the definition of success based on IOP values reported in the Methods section are not standardized and widely applicable in clinics; however, they have been used in several studies and are based on criteria reported in the World Glaucoma Association Guidelines published in 2009 [37].

The study adds to the very limited current literature in this field and could be of clinical importance to clinicians when managing post-surgical canaloplasty patients with PEXG. Our results may help pave the way to future studies regarding physiopathological mechanisms behind acute IOP spikes in these patients, which could be due to decreased outflow related to the effects of the prolene suture in Schlemm’s canal. This could be of importance in those eyes with existing compromised angular tissue structures because of the long-term effects of PEX deposits, in addition to the cumulative side effects of numerous years of local medication. Comparative prospective studies based on traditional canaloplasty and surgery involving viscodilation of Schlemm’s canal without the positioning of a prolene suture (i.e., ab interno canaloplasty) could be useful in clarifying the potential effects on outflow mechanisms.

## 5. Conclusions

Canaloplasty is a very interesting and fascinating surgical technique, which can offer good results, especially in POAG, juvenile and pigmentary glaucoma with very high IOP. The long-term outcomes in patients with PEXG may seem satisfactory at first glance, considering that canaloplasty can maintain post-operative IOP values at physiological values for numerous years in most cases. Unfortunately, our study showed that more than 60% of cases can develop an abrupt rise in IOP occurring several years after surgery. In some cases, these spikes can go unnoticed upon onset or be detected considerably late, leading to a potentially dramatic progression of the functional damage.

In order to limit the serious risks related to potential undetected IOP elevations after surgery, canaloplasty should either be avoided in PEXG eyes or only considered as a possible option in selected patients having a high risk of failure with filtrating surgery. These patients need to be carefully assessed after canaloplasty, even if it is apparently successful, and should be clearly informed about the advantages and potential risks of this surgical procedure. Moreover, patients need to be educated about the acute signs and symptoms of IOP spikes and be informed of the possible need for future filtrating surgery.

## Figures and Tables

**Figure 1 jcm-11-02532-f001:**
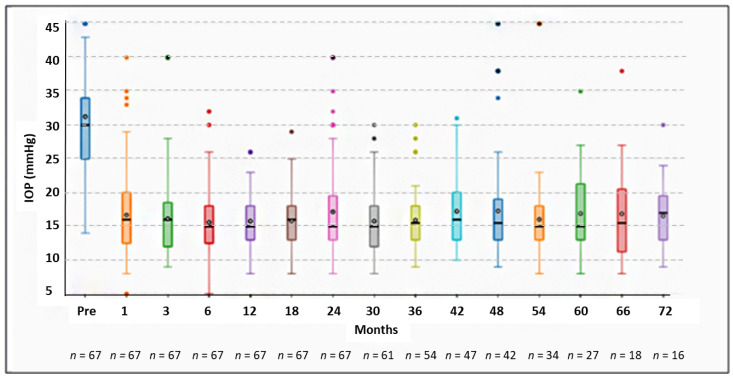
Box plot representation of IOP values over time in 7 years of follow-up in the cohort of 67 PEXG eyes that underwent canaloplasty.

**Figure 2 jcm-11-02532-f002:**
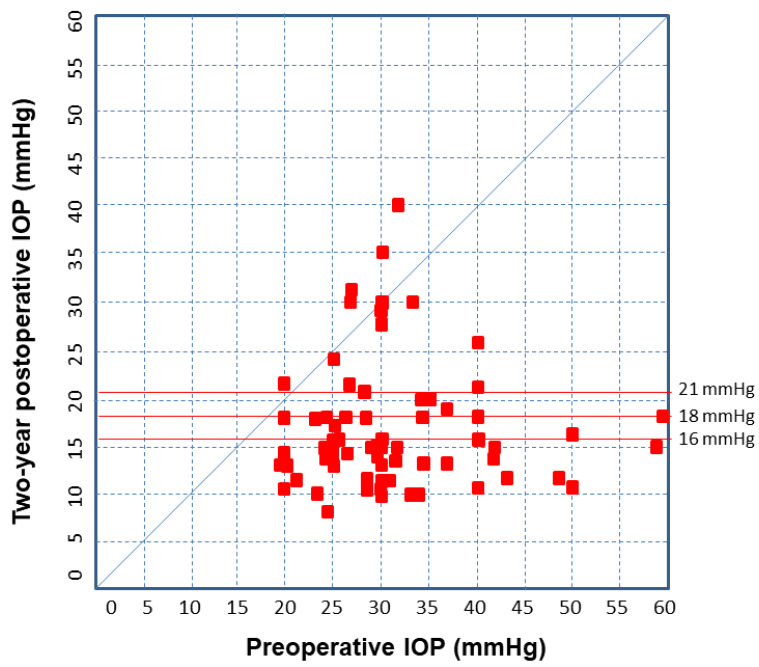
Scatter plot of IOP values before surgery and after canaloplasty in 67 PEXG eyes after 2 years.

**Figure 3 jcm-11-02532-f003:**
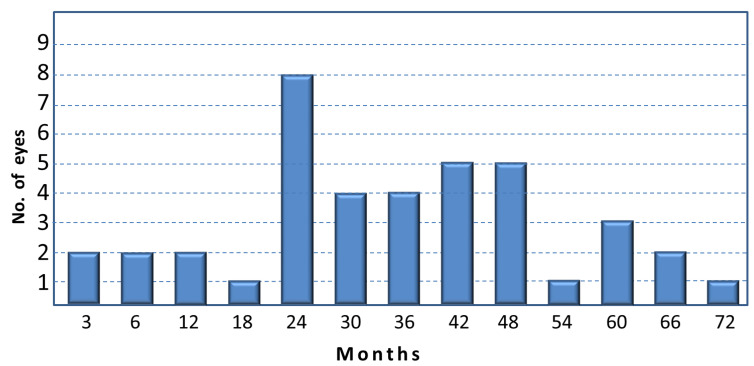
Number of eyes with an IOP increase >25 mmHg after canaloplasty.

**Table 1 jcm-11-02532-t001:** Qualified success rates after 2, 3 and 4 years.

	Post-operative IOP		
	≤ 21 mmHg	≤ 18 mmHg	≤ 16 mmHg
After 2 years (67 eyes)	54 (80.6%)	49 (73.1%)	41 (61.2%)
After 3 years (54 eyes)	50 (92.6%)	43 (79.6%)	31 (57.4%)
After 4 years (42 eyes)	35 (83.3%)	29 (69.0%)	23 (54.8%)

IOP: Intraocular pressure.

**Table 2 jcm-11-02532-t002:** Complete success rates after 2, 3 and 4 years.

	Post-operative IOP		
	≤ 21 mmHg	≤ 18 mmHg	≤ 16 mmHg
After 2 years (67 eyes)	28 (41.8%)	26 (38.8%)	24 (35.8%)
After 3 years (54 eyes)	22 (40.7%)	20 (37.0%)	14 (25.9%)
After 4 years (42 eyes)	9 (21.4%)	9 (21.4%)	9 (21.4%)

## Data Availability

Not applicable.

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
