# Peer review of "Canaloplasty in Pseudoexfoliation Glaucoma. Can It Still Be Considered a Good Choice?"

_jcm, 2022, doi:10.3390/jcm11092532_

Round 1

Reviewer 1 Report

Dear authors,

In this study, the authors tried to assess the long-term outcomes and efficacy of canaloplasty surgery in PEXG. This observational study was from 2008 to 2022. For its acuted rise of IOP in more than 60% of patients after a long period, canaloplasty may not be suitable in eyes with PEXG. It is interesting and provides some references for clinical work. However, I have some specific comments on it:

  1. The study lack of comparison group. So the conclusion was not suitable to estimate the efficacy of canaloplasty in PEXG, without comparison with other surgeries or with other diseases.
  2. In the “Materials and Methods” part, what does “qualified success” mean by saying “The definition of success was based on three different criteria…… with or without medical treatment (“qualified success”)”? The cut-off was 21mmHg, 18mmHg, and 16mmHg. Why did the author divide it like this? Were there any references?
  3. In the “Results” part, there were patients lost during the observational period. I advised the authors to conduct a survival analysis to study the relationship between outcome and survival time, and deal with missing data.
  4. In the “Results” part paragraph 1, what does it mean by saying “the mean IOP after 24 months was 17.2±6.7mmHg”? What was the exact time point (from when to when) when IOP was 17.2mmHg?
  5. Line 153-154: “The number of medications used pre- and at the 2, 3, and 4-year follow-up was 3.5±9, 1.2±1.4, and 1.3±1.3, and 1.9±1.3 respectively.” Please verify the p value between each group.
  6. Line 159-166: Please give more details about the follow-up treatment of complications.
  7. Figure 2 showed the scatter plot of IOP values before and after canaloplasty in 67 patients after 2 years. However, these 67 patients were followed up only for 2 years. Is there some mistake by saying “after”? Is “in the second year after operation” more appropriate?
  8. The study showed that more than 60% of the cases developed an abrupt rise of IOP occurring several years after surgery. How did the authors deal with these patients with elevated IOP and what were the results after treatment? Please describe and discuss further in the article.
  9. More details should be included in the demographic and basic clinical data of the enrolled patients, such as gender, glaucoma drug types, BCVA, cataract surgery or not, optic nerve, VF, et al.
  10. How many patients with late failure of IOP spike was defined failure at final follow-up? And how many of them had to perform secondary glaucoma surgeris?
  11. Discussion: personal unpublished data should not be the evidence for discussion. And all the discussion part lack of enough references.
  12. The conclusion part should be focused of what this study found, but not comparison with other studies, neither some statement of unrelated guesses. For example, “only considered as a possible option in patients with chronic conjunctival inflammation that may have a greater risk of filtering bleb failure with filtrating surgery.” The condition of conjunctiva was not analyzed in this study.

Reviewer 2 Report

Line – 21. Pl. correct mm/Hg to mmHg

Line 23 – criteria mentioned are IOP ≤ 21 mmHg, ≤18 mmHg and ≤16 mmHg. As per guidelines on design and reporting glaucoma surgical trials they are ≤ 21, ≤18  and ≤15. Pl. explain.

Line 45 – pl. check whether there is a spelling mistake in word athalamia?

Line 187 – pl. check the meaning of the sentence

Table 3 is absent in the text.

No number and description for figure – bar diagram is provided. Pl. provide the same.

The cumulative number of eyes in table 1 (qualified success rates……) and table 2 (complete success rates…..) exceeds total number of investigated eyes. For example, qualified success with criteria IOP ≤ 21 mmHg at 2 years was achieved in 54 eyes (table 1) and complete success in 28 eyes (table 2), making the sum of eyes to be 82 eyes, whereas there were only 67 eyes investigated. Pl. check your data and make the necessary changes. Or pl. explain the discrepancy.

Round 2

Reviewer 1 Report

None